# Current and Future Treatments in Primary Ciliary Dyskinesia

**DOI:** 10.3390/ijms22189834

**Published:** 2021-09-11

**Authors:** Tamara Paff, Heymut Omran, Kim G. Nielsen, Eric G. Haarman

**Affiliations:** 1Department of Paediatric Pulmonology, Emma Children’s Hospital, Amsterdam UMC, 1105 AZ Amsterdam, The Netherlands; t.paff@amsterdamumc.nl; 2Department of General Pediatrics, University Childrens’s Hospital Muenster, 48149 Muenster, Germany; heymut.omran@ukmuenster.de; 3Danish PCD Centre, Danish Paediatric Pulmonary Service, Department of Paediatrics and Adolescent Medicine, Righospitalet, Copenhagen University Hospital, DK-2100 Copenhagen, Denmark; kim.g.nielsen@regionh.dk; 4Department of Clinical Medicine, University of Copenhagen, DK-2100 Copenhagen, Denmark

**Keywords:** primary ciliary dyskinesia, treatment, genetic

## Abstract

Primary ciliary dyskinesia (PCD) is a rare genetic ciliopathy in which mucociliary clearance is disturbed by the abnormal motion of cilia or there is a severe reduction in the generation of multiple motile cilia. Lung damage ensues due to recurrent airway infections, sometimes even resulting in respiratory failure. So far, no causative treatment is available and treatment efforts are primarily aimed at improving mucociliary clearance and early treatment of bacterial airway infections. Treatment guidelines are largely based on cystic fibrosis (CF) guidelines, as few studies have been performed on PCD. In this review, we give a detailed overview of the clinical studies performed investigating PCD to date, including three trials and several case reports. In addition, we explore precision medicine approaches in PCD, including gene therapy, mRNA transcript and read-through therapy.

## 1. Primary Ciliary Dyskinesia

Primary ciliary dyskinesia (PCD) is a rare, mainly respiratory, ciliopathy that is characterized by recurrent upper and lower respiratory tract infections due to abnormal function of the cilia lining the respiratory epithelium. In most cases, PCD is an autosomal recessive disorder. However, more recently, x-chromosomal and autosomal dominant modes of inheritance were described [1,2,3,4,5]. The disturbed mucus clearance results from immotile or abnormal motion (dyskinetic) of the respiratory cilia lining the upper and lower airways. In some cases, there is a severe reduction in the number of respiratory cilia, though individual cilia may show normal motion [6], thus contradicting the term PCD. Therefore, primary respiratory ciliopathy is a more accurate term for this spectrum of diseases. Interestingly, the determination of left–right lateralization during embryogenesis is determined by the motility of the nodal monocilia, which have many ultrastructural similarities with respiratory cilia. Consequently, up to half of all patients with PCD have a situs inversus or situs ambiguous due to randomization of the left–right body asymmetry [7]. Similarly, the structure of the flagella on sperm cells and the cilia lining of the fallopian tubes and vasa deferentia share many components with respiratory cilia, and male (often) and female (less frequently) infertility can occur [8,9]. The reduced generation of multiple motile cilia was associated with hydrocephalus, probably due to dysfunction of ependymal cilia motility [5,10]. Though recurrent respiratory tract infections eventually result in progressive lung damage and bronchiectasis, it was generally assumed that the disease course in PCD was relatively benign compared to cystic fibrosis and did not result in a shortened lifespan. However, several genotypes were identified, including mutations in *CCDC39*, *CCDC40*, *MCIDAS* and *CCNO*, with a much more aggressive disease course [6,11,12,13,14]. In particular, in patients with *CCNO* mutations, lung damage progresses rapidly and lung failure frequently ensues, necessitating lung transplantation [14]. On average, patients with PCD lose approximately 0.8% FEV_1_ per year [15]. In addition, the disease burden is high and the quality of life is low, especially in adults [16,17].

So far, the restoration of ciliary function has not been possible in patients and treatment has been focused on improving mucociliary clearance (physiotherapy, mucolytics) and the early treatment of respiratory tract infections [18]. Though some studies indicate that with early diagnosis and the start of treatment, the disease course is more favorable [19], this was not replicated by others [15]. Therefore, there is an urgent need for evidence-based and effective treatments for this group of patients, including treatments that directly aim at improving or normalizing ciliary function. In this review, we summarize the data that is available so far and explore several precision medicine approaches, including gene therapy, mRNA transcripts and read-through therapy.

## 2. Current Treatments in Primary Ciliary Dyskinesia

As there are no curative options yet, PCD treatment is directed at preventing and managing disease complications. With only three randomized controlled trials in PCD so far, treatment guidelines are mainly based on expert opinion and extrapolations from CF and non-CF bronchiectasis guidelines, despite having distinct pathophysiologies [20,21,22,23]. As a result of the paucity of evidence-based therapeutic options, treatment approaches have varied widely between European countries [24]. Through the EU projects BESTCILIA and BEAT-PCD, international collaborative efforts recently led to the iPCD Cohort and the international PCD registry [25,26]. These important studies provide a more detailed characterization of the disease course and will guide patient diagnosis and treatment, as well as help identify eligible patients for RCTs [26,27,28]. Such international platforms, together with improved diagnoses in adult patients and engagement with patient support groups, are key in developing and testing current and novel treatment approaches in PCD [25,29,30]. To this end, a clinical trials network was established within ERN-LUNG (CTN-PCD), including 22 clinical trial sites in 12 countries in Europe and the United Kingdom and over 1200 adult and 1400 pediatric individuals with PCD so far.

### 2.1. Airway Clearance Therapy

Mucous retention promotes bacterial infections and inflammation, which in turn can lead to pulmonary exacerbations and bronchiectasis. As a result, quality of life, lung function and structure are relatively poor in PCD patients [16,31,32]. Treatment options that are used to improve the hampered mucociliary clearance in PCD include the use of mucolytics, hyperosmolar agents and physiotherapy applying airway clearance techniques [24].

#### 2.1.1. N-Acetylcystein

Mucolytic agents target increased sputum viscosity to make sputum easier to clear. N-acetylcystein (NAC) can decrease mucus viscosity in vitro by depolymerizing the mucin glycoprotein oligomers and displays anti-inflammatory and antioxidant activity. Inhaled NAC is proposed to act primarily through its mucolytic effects but is usually not tolerated well, as the low pH often induces cough and bronchospasm. Given orally, it is unlikely to have a direct mucolytic effect, as it is not detectable in the airways after administration [33,34,35,36]. However, it is thought to act by reducing oxidative stress and inflammation. Long-term high-dose oral NAC was shown to reduce the exacerbation rate and improve quality of life in adult COPD and non-CF bronchiectasis patients [37,38]. The first-ever double-blind, placebo-controlled, cross-over trial in PCD patients tested the effect of NAC in 13 PCD patients and 44 CF patients after three months of 200 mg three times daily (for patient weight < 30 kg) or 400 mg dosing twice daily (for patient weight >30 kg) [21]. Although CF patients showed improvement in lung function, in PCD patients, there was no improvement in the subjective clinical scores or lung function. However, the study was hampered by its small sample size and lack of a wash-out period between the interventions. In CF, the use of NAC remains a matter of debate as good-quality studies are lacking [39]. However, there are some recent encouraging data to support its use. A phase 2 randomized placebo-controlled trial using 70 CF patients showed a stable-to-improved lung function in the treatment group after 24 weeks compared to a decline in lung function in the placebo group [40]. 

#### 2.1.2. Hypertonic Saline

Inhaled hyperosmolar agents improve airway clearance by hydrating the viscous airway secretions and by stimulating cough [41,42,43,44]. In CF, nebulized hypertonic saline (HS) reduces the frequency of pulmonary exacerbations in children >12 years and adults, and slightly improves the quality of life of adult patients [45]. In non-CF bronchiectasis, the evidence is contradictory, but a well-designed 52-week trial on the effect of HS on exacerbation frequency in adults is underway [42,46,47,48]. A carefully designed double-blind, randomized, placebo-controlled cross-over study in PCD evaluated the effect of inhaled HS compared to inhaled isotonic saline (IS) on quality of life in 20 adult PCD patients [22]. This was the first randomized double-blind RCT to include only patients with confirmed PCD according to international diagnostic standards [49]. Further, it concealed allocation from patients, investigators and treating physicians, and the taste of hypertonic saline was masked with quinine sulphate. The use of HS was safe and well tolerated. Although the study showed a minor increase in the primary outcome measure quality of life total score using SGRQ after 12 weeks of daily HS inhalation compared to IS inhalations, the difference did not reach the minimal clinically important difference or statistical significance. However, a small but clinically and statistically significant improvement on the QoL-B Health Perception scale was observed after the hypertonic saline treatment. Further, clinically important improvements after hypertonic saline treatment in the QoL-B Respiratory Symptoms and Vitality scales were observed, but they did not reach statistical significance. The change in the Role Functioning scale was borderline significantly different between the treatment and control phase, but the effect size was below the MCID.

Although the study nearly reached the target sample size, it was underpowered as the variability in outcome measures was larger than anticipated. Unfortunately, based on the negative findings on the primary outcome measure, several insurance companies have concluded that hypertonic saline is not effective in PCD and do not reimburse this medication [29]. Further, PCD HR-QOL did not exist at the time. Thus, it seems likely that patients can benefit from HS treatment, but larger sufficiently powered studies are urgently needed. Furthermore, the administration of HS via a face mask may be more effective by additionally addressing upper airway problems in PCD and the clinical outcome measures used may not have been sensitive enough. Detailed data on the variability observed will help design future studies.

More recently, the CLEAN-PCD study was performed (https://clinicaltrials.gov/ct2/show/NCT02871778, accessed on 8 September 2021). The CLEAN-PCD trial is an international phase 2 cross-over RCT that aims to test the safety and the effect on the quality of life of an epithelial sodium channel (ENaC) inhibitor (VX-371) with and without Ivacaftor, a CFTR potentiator, or hypertonic saline. ENaC inhibitors are thought to improve airway hydration and mucociliary clearance by blocking the reabsorption of sodium in the airway surface liquid. The results of this study are currently being produced.

#### 2.1.3. Recombinant Human Dornase Alfa

Dornase alfa is a highly purified solution of recombinant human deoxyribonuclease I (rhDNase), which is an enzyme that selectively cleaves DNA. When inhaled, it hydrolyzes the DNA in purulent sputum and thereby improves the sputum’s viscoelasticity and clearability from the airways. rhDNase is highly effective in improving lung function and decreasing pulmonary exacerbations in CF [50]. However, in patients with non-CF bronchiectasis, inhalations were found to be ineffective and potentially harmful in two separate studies [51,52]. Daily use of rhDNase increased exacerbations and decreased lung function in these subjects and this study underlines the risk of extrapolating treatments from CF to the very heterogeneous group of non-CF bronchiectasis with distinct pathophysiologic mechanisms, one of which is PCD. Although the use of rhDNase is currently not recommended in PCD expert guidelines because of the lack of clinical trials in PCD patients, there are several anecdotal case reports in children with PCD that show a vast improvement in respiratory symptoms and lung function following the start of rhDNAse treatment and a decline after discontinuation [53,54,55]. A 14-year-old girl with PCD and gastrointestinal reflux experienced a decrease in sputum volume and cough <72 h after the start of treatment and a 20% improvement in FEV_1_ after 4 weeks, which were maintained after 4 months. In a term neonate with PCD suffering from respiratory distress, dyspnea subsided, oxygen saturation improved and breath sounds normalized within a few days after starting on rhDNase therapy, which was continued until the age of 7 months. A worsening of symptoms and admittance followed after two periods of treatment withdrawal with direct improvement after the resumption of rhDNase treatment. Two PCD siblings of 6 and 17 years who were started on rhDNase both showed improvement in respiratory symptoms and lung function and discontinuation of treatment was followed by a decline in lung function. These reports could reflect a positive effect on PCD patients and rhDNase treatment should be evaluated in an RCT.

### 2.2. Antibiotic Treatment

The cornerstone of PCD management remains microbial surveillance through sputum cultures, oropharyngeal cough swabs or laryngeal suction and treatment of acute pulmonary exacerbations with oral or intravenous antibiotics, depending on the severity of the clinical symptoms [20]. Recently, an international BEAT-PCD consensus statement for infection prevention and control was published [56]. Airway secretion samples should be cultured at least four times annually on selective media for *Pseudomonas aeruginosa* and annually for *nontuberculous mycobacteria* (NTM). Antibiotic treatment is based on the type of bacterial pathogen and whether a patient is symptomatic. The consensus statement advises to treat a patient with a positive culture for *Pseudomonas aeruginosa*, *methicillin-resistant Staphylococcus aureus* (MRSA) or *Burkholderia cepacia* complex, regardless of symptoms. There is a general concern for chronic *Pseudomonas aeruginosa* infection in PCD as a substantial prevalence is reported in several cohorts [57,58,59,60]. Chronically infected patients are at risk for decline in lung function and deterioration in structural lung changes [31,61]. There is no evidence on how to manage *Pseudomonas aeruginosa* infection effectively in PCD in terms of eradication, chronic infection or pulmonary exacerbation. Consequently, considerable variation in the management of PCD is found across European centers. Intravenous, inhaled and oral antibiotics are used singly or in combination [62]. Irrespective of this variability, it is important to recognize that *P. aeruginosa* colonizes both the upper and lower airways in PCD patients and that eradication treatment should thus not only be aimed at the lungs [63].

There is no universally accepted definition of exacerbations in PCD for clinical practice. Recently, a consensus statement on pulmonary exacerbations in PCD for clinical trials was published. An exacerbation is defined by the presence of three or more of the following seven items: (1) increased cough; (2) change in sputum volume and/or color; (3) increased shortness of breath, as perceived by the patient or parent; (4) decision to start or change antibiotic treatment because of perceived pulmonary symptoms; (5) malaise, tiredness, fatigue or lethargy; (6) new or increased hemoptysis; and (7) temperature >38 °C [64]. Antibiotic treatment is guided by culture history and microbial sensitivity from upper and lower airway secretions. Despite the lack of evidence, a total duration of therapy of 14–21 days is recommended to be performed in patients with CF and non-CF bronchiectasis [65,66].

#### Azithromycin Maintenance Therapy

Macrolides are known to have bacteriostatic properties, as well as anti-inflammatory and immunomodulatory effects. By inhibiting quorum sensing, they may protect against biofilm growth, which is an important mechanism characterizing some opportunistic bacteria, such as *Pseudomonas aeruginosa* [67]. Prolonged macrolide use was first reported in diffuse panbronchiolitis to significantly reduce mortality [68]. Since then, it has been researched in many other chronic respiratory suppurative diseases that are dominated by neutrophilic inflammation. In CF, there seems to be a consistent improvement in FEV_1_ (mean difference at six months of 3.97% (95% confidence interval = 1.74 to 6.19%; *n* = 549, from *n* = 4 studies) and pulmonary exacerbations (odds ratio = 1.96 (95% confidence interval = 1.15 to 3.33)) [69,70]. Two studies in adult non-CF bronchiectasis lasting 6 and 12 months showed similar results [71,72]. For a long time, there were only anecdotal reports of benefit in PCD patients [73,74]. The recently performed BESTCILIA multicenter phase 3 trial investigated the effect of 6 months azithromycin on exacerbation frequency in 90 PCD patients [23]. Subjects were randomly assigned to use azithromycin 3 times a week or a placebo. Azithromycin was well tolerated and halved the rate of exacerbations (rate ratio = 0.45, 95% CI = 0.26–0.78; *p* = 0·004). Furthermore, the rate of detected pathogenic bacterial species was significantly lower in the azithromycin group compared with the placebo group (RR = 0.34, *p* < 0.0001). Although macrolide resistance should be considered an important risk, macrolide-resistant bacteria can be eradicated with other commonly used antibiotics. This well-designed study is the first to enroll a large number of PCD patients and to show a dramatic improvement in pulmonary exacerbations. This underlines the crucial role of collaborative networks in conveying such important trials.

### 2.3. Bronchodilators and Inhalation Corticosteroids 

Bronchodilators and inhaled corticosteroids (ICS) are currently only recommended in PCD patients with co-existing asthma [20]. In practice, they are often prescribed to PCD patients who present with a recurrent wheeze [24,75]. Dehlink and co-workers reported that 35% of their PCD patients were prescribed high dose ICS (median = 500 mg) of which two-thirds was in a fixed-dose combination with long-acting beta agonists (LABA) [75]. Half of the patients showed signs of atopy, about one-third of PCD patients had reversible airway obstruction >12% and their FeNO was typically low. Hellinckx and co-workers described bronchial obstruction in their cohort of 11 PCD patients, which was in part reversible after using salbutamol (mean reversibility 13,2% FEV1) [76]. 

In contrast to asthma patients, PCD patients showed neutrophilic airway inflammation, and reversible airflow obstruction may also be caused by mucus shifting. This is supported by a report showing that exercise causes greater bronchodilation than a short-acting B2 agonist [77]. However, additional bronchodilation due to short-acting B2 agonists may theoretically also improve sputum evacuation in patients with PCD. Their role in the management of PCD should thus be further investigated. 

In CF patients, ICS have also been widely used in the past but are no longer recommended, as evidence for its benefit is lacking [78]. RCTs are needed to determine whether ICS have a role outside eosinophilic inflammation, specifically in PCD patients. 

## 3. Ear, Nose and Throat (ENT) Treatment

### 3.1. Sinonasal Disease

The majority of PCD patients suffer from chronic rhinosinusitis [79,80]. Chronic rhinosinusitis management in PCD focuses on the relieving symptoms of nasal obstruction and discharge and is extrapolated from treatment guidelines in the general population. It may include sinonasal irrigation with iso- or hypertonic saline, topical nasal steroids and antibiotics. In addition, a few studies showed a benefit of functional endoscopic sinus surgery in PCD [20,81,82]. However, anatomical sinonasal anomalies, such as aplasia or hypoplasia of the sinuses, particularly of the frontal and sphenoid sinuses, were also described [79,83]. Nasal polyps may occur in up to half of all patients and may be another indication for surgery [84,85]. 

### 3.2. Otologic Disease

Otologic manifestations in PCD are frequent and mainly include chronic secretory otitis media with frequent superinfections, resulting in otitis media acuta (OMA). Hearing loss is reported in up to half of all patients [86,87]. The majority of patients with hearing loss suffer from altered air conduction (conductive hearing loss) and the prevalence of sensineuronal hearing loss increases with age [85,88,89]. When hearing loss happens early in life, it may be complicated by a delay in speech development [90]. Hearing aids are therefore recommended in children with moderate-to-profound hearing loss. In cases where there is otorrhea following tympanic perforation or due to insertion of ventilation tubes, bacterial surveillance should direct antibiotic therapy. The use of ventilation tubes in children with PCD is controversial because recurrent or prolonged purulent ear discharge was described in up to half of all patients [89,91,92]. Despite the high prevalence, knowledge of ear, nose and throat (ENT) problems in PCD patients is limited. The Ear-Nose-Throat Prospective International Cohort of PCD patients (EPIC-PCD) aims to shed more light on this matter by characterizing ENT disease in PCD patients and evaluating its association with lower respiratory disease.

In summary, due to the lack of evidence-based therapeutic options, treatment guidelines in PCD are currently mainly based on expert opinion and extrapolations from CF and non-CF bronchiectasis care. The latter induces a huge risk of exposing PCD patients to ineffective or even harmful therapies and depriving them of possible beneficial treatments. As some previous studies suffered from recruitment problems, the use of insensitive outcome measures and inter-patient variability, there is a huge need for international collaboration, detailed patient characterization and the involvement of support groups to perform essential therapeutic trials in PCD patients. The BESTCILIA trial with azithromycin maintenance therapy can be considered a milestone in PCD treatment, incorporating all these aspects [23]. As part of ERN-LUNG, PCD-CTN provides a crucial platform for future clinical trials.

## 4. Future Treatments in Primary Ciliary Dyskinesia

So far, treatments have focused on symptomatic relief by reducing sputum viscosity and providing early treatment of infections. However, as described above, early initiation of treatment has not yet resulted in an unequivocal stabilization or improvement of the disease course. The identification of specific genotypes in PCD and underlying disease mechanisms was the first step toward personalized medicine, with the ultimate goal of restoring ciliary function. Gene therapy encompasses treatment strategies that either replace (classical gene therapy) or ‘repair’ the mutated gene sequences (gene editing). As the conducting airways are relatively accessible to the administration of a vector, gene therapy may be a viable option for PCD. To date, three studies have been published describing the partial restoration of ciliary function in ciliopathies using classical gene therapy and one study using gene editing.

### 4.1. Gene Therapy

The first study was done by Chhin and co-workers in 2009 [93]. In this study, the authors tried to restore ciliary beating in DNAI1-deficient human airway epithelial cells in vitro. The *DNAI1* gene encodes an important component of the outer dynein arm (ODA), namely, the axonemal dynein intermediate chain protein. Dynein arms are the molecular motors that are responsible for microtubule doublets sliding in the axoneme. Mutations in *DNAI1* account for approximately 10–14% of the cases in PCD [94].

A lentiviral vector containing cDNA of *DNAI1* driven by the elongation factor 1 promoter was used. The researchers chose a lentiviral vector, as it can integrate its genetic material into the host cell genome of non-replicating cells and it is relatively weakly immunogenic. The transduced *DNAI1* was transcribed and translated into protein, where the outer dynein arms reappeared, as demonstrated by TEM, and ciliary motility was partially restored. As up to half of the infected cells were not transduced, epithelial cell aggregates were observed with mosaics of cells with motile and immotile cilia. A partially uncoordinated beating pattern was thus observed. 

The second study was done by Ostrowski and co-workers [95]. They used a mouse model carrying an inducible deletion coding for the intermediate dynein chain *Dnaic1*, which is the murine homolog of *DNAI1* (82% identical). Treatment of these mice with tamoxifen resulted in the activation of tamoxifen-inducible Cre-recombinase and the deletion of two loxP-flanked exon sites in *Dnaic1* [96]. In this manner, adult mice with severely impaired mucociliary clearance in the nasopharynx and trachea were developed without the usually occurring lethal hydrocephalus. Using a lentiviral vector pseudotyped with avian influenza hemagglutinin, gene transfer to undifferentiated and differentiated cultures of mouse *Dnaic^-/-^* cells partially restored Dnaic1 expression and ciliary activity. The vector thus transduced the airway epithelium via the apical surface, which is a major obstacle in previous gene therapy studies of lung diseases. The ciliary beat frequency did not differ between positive controls and transduced cells. However, the percentage of corrected cells was only 10–15% of the positive control cultures. Though this seems low, the authors demonstrated using their inducible mouse model with varying doses of Tamoxifen that only 20% of mucociliary clearance activity was required to prevent rhinosinusitis in their mouse model. On the other hand, they also demonstrated that rhinosinusitis severely hampered the gene transfer of a β-galactosidase-expressing vector in *Dnaic1^−/−^* mice. No in vivo data was provided on the restoration of ciliary function.

McIntyre and co-workers were the first to restore ciliary structure and function in vivo in a ciliopathy mouse model of ORPK-mice (Oak Ridge polycystic kidney disease) [97]. In this mouse model, the assembly of both motile and immotile cilia was disturbed due to a hypomorphic mutation in *IFT88* coding for an intraflagellar transport protein. The study focused on the primary (immotile) cilia lining the olfactory sensory neurons, which do not have a mucociliary clearance but instead have a sensory function. In mutated mice, there was a severe reduction in cilia. The remaining cilia were shortened and malformed. Olfactory function was severely reduced, as determined both functionally (electro-olfactogram) and biochemically (S100a5 expression). To restore the ORPK phenotype, mice were injected intranasally with the *IFT88*-GFP adenovirus. Interestingly, in transfected cells, ciliary expression and function were restored. In addition, there was a clear improvement in odor-guided feeding behavior. The authors argued that given the fact that olfactory epithelium undergoes constitutive neurogenesis throughout life, the duration of expression will probably be limited by the lifespan of the neurons and, thus, that stable incorporation in the olfactory sensory neurons is not necessary. On the other hand, stable transfection of olfactory epithelial stem cells may circumvent the need for repeated treatments. The effects on motile cilia were not studied. 

The approaches described above (classical gene therapy) are very challenging regarding many of the PCD genes, as they are very large in size and therefore difficult to be transported by current vectors. In addition, transferred genes are driven by different promotors than wild-type genes, and the physiological regulation of production is different. This may have negative consequences for the protein function and interaction within the protein machinery in epithelial cells. Third, in many cases, the transferred gene is permanently integrated into the host genome with possible serious side effects. In the French and British SCID-X1 trials, where patients with congenital severe combined immune deficiency syndrome were treated with gammaretroviral gene therapy, 5 out of 20 patients developed leukemia [98]. This is possibly related to preferential insertion in the 5′ ends of *dangerous* genes, such as signal transduction, proliferation and proto-oncogenes.

Gene editing, which is a technique that replaces mutated wild-type sequences, is not affected by gene size. The key to genome editing is creating site-specific double-strand breaks with engineered nucleases, followed by homologous recombination in the presence of the homologous DNA segments. To date, there are three distinct classes of site-specific endonucleases that have been bioengineered/discovered: zinc finger nucleases (ZFNs), transcription-activator-like effector nucleases (TALEN) and the clustered regularly interspaced short palindromic repeats (CRISPR/Cas9). DNA-binding specificity is higher, off-target effects are lower and construction of the DNA-binding domains is easier in CRISPR/Cas9 compared to TALEN and especially ZFNs. However, since CRISPR/Cas9 is only recently available, no published data on ciliopathies are available yet. Off-target effects were described in all approaches, including CRISPR/Cas9-mediated gene transfer.

The first and so far only study applying gene editing in PCD was done by Lai and co-workers and was published in 2016 [99]. They aimed to restore ciliary function in vitro in epithelial cell lines (293T cells) containing the mutated *DNAH11* target site and in epithelial cell cultures obtained from two PCD patients with *DNAH11* nonsense mutations using TALEN. *DNAH11,* which accounts for approximately 6% of the cases in PCD, is a component of the dynein heavy chain in the outer dynein arm and is essential for normal ciliary motility [100,101,102]. This rather large gene is 353 kb in size and encompasses 82 exons, resulting in a 14 kb mRNA [103]. This is too large for most currently used delivery systems. TALEN cleaved 80% of the mutated *DNAH11* in the epithelial cell line and replaced the mutated sequence in approximately 50% of the cells. In epithelial cell cultures derived from the PCD patients, site-specific recombination was observed in 33% of the cells and normalization of ciliary beating in 29%. No frameshifts were introduced. This study shows that gene editing can partially restore ciliary activity ex vivo in *DNAH11* mutant epithelial cells.

These studies showed that the restoration of ciliary activity using gene therapy is possible. However, multiple challenges to successful therapy were also identified, including the resistance of differentiated airway epithelium to apical transduction by viral vectors, obtaining sufficiently high transfection rates to restore ciliary activity, the inhibition of gene transfer by airway secretions and difficulties in obtaining long-term expression. One must also consider the development of immune responses to the proteins produced after successful transfection or the vector components. In addition, there are safety concerns about possible lethal side effects due to off-target effects, as was demonstrated by an early gene therapy trial in congenital immunodeficiency diseases [98].

### 4.2. Transcript Therapy

A further downstream treatment approach is so-called transcript therapy or RNA therapy. This approach has major advantages in that it does not induce any alterations in genomic DNA, effects are reversible and, consequently, this approach has no malignant potential. RNA therapies can be sorted into three categories: those that encode proteins, those that target proteins and those that target nucleic acids.

Recently, during the ATS 2021, data from the company TRANSLATE-BIO was presented by Woo and co-workers on mRNAs encoding the sequence of human *DNAI1* packaged in proprietary lipid nanoparticles (LNPs). The *DNAI1*-LNPs were administered to mouse lungs and DNAI1 knockout human induced pluripotent stem cells (iPSCs) in air–liquid interface (ALI) cultures. The DNAI1-LNP showed expression when delivered directly to the lungs of mice using either the intratracheal or nebulized route of administration. *DNAI1* expression was detected using Western blot and localization in bronchiolar multi-ciliated cells was demonstrated using immunohistochemistry. No data was provided on the restoration of cilia function in iPSCs.

Another example of a transcript therapy encoding proteins in PCD is being explored by ETHRIS (Planegg, Germany) in collaboration with the research group of Heymut Omran. ETHRIS developed ETH42, which is based on their SNIM^®^RNA (Stabilized Non-Immunogenic mRNA) technology platform. ETH42 provides full-length mRNA instructions for synthesizing the CCDC40 protein (https://ethris.com/eth42-in-pcd/, accessed on 8 September 2021). Mutations in *CCDC40* occur in approximately 10% of PCD patients and represent one of the worst phenotypes with rapid deterioration of lung function and low BMI. The studies are still in the pre-clinical stage, but the first results appear promising.

Therapies targeting nucleic acids can be subdivided into single-stranded antisense oligonucleotides (ASO’s) and double-stranded molecules. The latter operate through RNA interference (RNAi). ASOs are short, modified DNA molecules consisting of 13–25 nucleotides. ASOs can prevent mRNA from being translated or alter mRNA splicing. One of the first ASOs to be approved by the FDA was eteplirsen for the treatment of Duchenne muscular dystrophy [104]. This ASO is an example of an exon-skipping drug that blocks the mutated portion of the gene from being expressed, resulting in the expression of a protein that is functional. So far, no data on these therapies have been published or reported regarding PCD.

### 4.3. Read-Through Therapy

In up to 28% of patients with PCD, one or two nonsense mutations were identified, resulting in a premature termination codon (PTC) [102]. Due to an early stop codon, translation into protein is prematurely terminated. The resulting protein is often non-functional and sometimes even harmful to the cell. Given the relatively high prevalence of nonsense mutations in genetic diseases and the severity of the phenotype, many efforts have been made to identify PTC therapeutics.

In the past, several small molecules were identified, including aminoglycosides or aminoglycoside analogs Ataluren and ELX-02 that bind to the decoding center of the ribosome and affect its translation fidelity. As a result, near-cognate and sometimes the original amino acid is incorporated into the polypeptide chain. Over many translations, this results in various protein variants with varying levels of activity. An important possible side-effect of nonsense suppression therapy is the possibility of natural termination codon read-through, resulting in the disruption of cellular processes.

So far, one molecule has been approved for clinical use: Ataluren or PTC124 for patients with Duchenne muscular dystrophy [105]. Studies in cystic fibrosis with PTC124 were disappointing, though a notable treatment effect was observed in individuals not receiving inhaled tobramycin [106]. ELX-02 was more recently described and shows promise based on in vitro data in human-derived intestinal organoids with G542X nonsense mutations and appeared safe based on phase 1 studies in healthy volunteers [107]. Currently, in vitro studies on PCD are being performed by Eloxx Pharmaceuticals (Waltham, United States of America, in collaboration with University Medical Center Utrecht (The Netherlands) and Amsterdam University Medical Center (The Netherlands). 

### 4.4. Drugs with Stimulatory Properties

Normal cilia beat in a highly coordinated and synchronized fashion. The ciliary beat frequency is dependent on dynein ATPase activity and the ability of cilia to increase it in response to various stimuli. Agents that were shown to increase CBF are beta 2 agonists, cholinergic drugs, adenosine triphosphate and phosphodiesterase inhibitors, mostly through a Ca^2+^/cAMP-dependent manner [108]. Based on the pathophysiology of PCD, where mucociliary clearance is disturbed due to the abnormal motion of cilia because of a structural defect, these drugs are not expected to have a positive effect on mucociliary clearance. However, with partial restoration of ciliary motility using one or more of the above-described treatments, mucociliary clearance may be improved by these drugs and, consequently, symptoms may be reduced.

## 5. Summary

PCD is a rare genetic disease in which mucociliary clearance is disturbed by the abnormal motion of cilia or a severe reduction in generation of multiple motile cilia, leading to recurrent upper and lower respiratory infections and, consequently, bronchiectasis. Treatment is aimed at (1) improving mucociliary clearance using physiotherapy and nebulization with mucolytics and (2) the eradication or reduction of upper and lower respiratory tract infections using early and aggressive antibiotic treatment, guided by microbial surveillance. Treatment guidelines are largely based on CF guidelines, as few studies have been performed on PCD. An unequivocal beneficial effect was established due to azithromycin maintenance therapy and, as such, this is the only evidence-based treatment found so far [23]. Therefore, additional trials are urgently needed to determine the effectiveness of different mucolytics (hypertonic saline, rhDNase), antibiotic eradication and maintenance regimes and ENT treatments. To coordinate these future studies, the PCD-CTN clinical trial network was established (https://ern-lung.eu/, accessed on 8 September 2021). The first results of these studies showing restored ciliary function in vitro using gene or transcript therapy are encouraging, but many hurdles still must be overcome before such treatments can be applied in a clinical setting. In conclusion, the first steps toward evidence-based treatment in primary ciliary dyskinesia were taken and encouraging in vitro data on restoring ciliary function are available, indicating that it is likely that during the next decade, major advances in the treatment of PCD are to be expected.

## Data Availability

Not applicable.

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
