# Peer review of "Current and Future Treatments in Primary Ciliary Dyskinesia"

_ijms, 2021, doi:10.3390/ijms22189834_

Round 1
Reviewer 1 Report
REVIEW
Current and future treatments in primary ciliary dyskinesia
This manuscript makes a very useful and systematic review of treatments for patients with PCD.
In the section on antibiotic treatment, I believe that it should also describe its adaptation to the microbiology of secretions, both from the upper and lower airways. Likewise, if the patient shows pseudomonas aeruginosa infection, what is the role of inhaled antibiotics ¿ (through the nose and mouth). Knowing the microbiology of upper and lower tract secretions and their coincidence or not is a critical point to approach the antibiotic treatment of these patients. The antibiotic treatment should be adapted to the patient's clinic ?. Is it the same in children as in adults? Do we use a single antibiotic or are we changing it over time?
In the section on the treatment of sinonasal disease, the authors should make it clear that hypoplasia or aplasia of the paranasal sinuses makes surgery on those sinuses impossible. Polyps are another thing, and they can be operated on without difficulty in their endonasal extension.
In the section on otological disease, it should be noted that secretory otitis media is the norm in PCD patientes and its superinfection leads to acute otitis media. Otorrhea occurs only when a tympanic perforation occurs (a rare occurrence) or when a transtympanic drain is placed. So, in many patients we do not have a discharge whose microbiology we can determine. We assume that the same bacteria participate in the infection as those in the upper airways ¿.
Section 4: Future treatments in primary ciliary dyskinesia is very interesting. However, its application in clinical practice seems distant in time
Authors should include general conclusions at the end of the manuscript regarding the current treatment and future perspectives of PCD.
Reviewer 2 Report
Overall, a timely review of the current status of therapy for PCD patients. The writing is informative and up to date. I do not have any concerns and recommend publication.
Round 2
Reviewer 1 Report
I am grateful to the authors who have made the modifications that I had proposed. I believe that the manuscript in its current form can be published.